# Prevalence and Factors Associated with Poor Respiratory Function among Firefighters Exposed to Wildfire Smoke

**DOI:** 10.3390/ijerph19148492

**Published:** 2022-07-12

**Authors:** Catarina Ramos, Beatriz Minghelli

**Affiliations:** 1Escola Superior de Saúde Jean Piaget Algarve, Instituto Piaget, 8300-025 Silves, Portugal; 21885@ipiaget.pt; 2Research in Education and Community Intervention (RECI), 3515-776 Viseu, Portugal

**Keywords:** firefighters, spirometry, respiratory function, wildfire smoke

## Abstract

One of the world’s biggest disasters are wildfires. The firefighting environment involves physical and respiratory risks, due the inhalation of fire smoke. This study aims to determine the respiratory function of firefighters exposed to wildfire smoke and explore the potential risk factors associated with poor respiratory function. The sample involved 53 firefighters, aged between 23 and 60 years (39.28 ± 8.71), 41 (77.40%) male and 12 (22.60%) female, who fought in wildfires. The measurement instruments used were as follows: a scale, a stadiometer, a questionnaire, a Fagerstrom test and a spirometer. Thirty-six (67.9%) firefighters showed a restrictive pattern. Firefighters fought between 1 and 9 (3.64 ± 1.97) fires and in total between 5 and 212 (62.34 ± 46.89) h. Nineteen (52.8%) firefighters, who showed a restrictive pattern, did not perform any physical exercise (*p* = 0.045). Twelve (70.6%) firefighters who practiced exercise and revealed a restrictive pattern trained at least 3 or less hours weekly, whilst five (29.4%) practiced more than 3 h (*p* = 0.030) of weekly exercise. Twenty (55.6%) firefighters with a restrictive pattern spent more than 48 h in combat (*p* = 0.029) and twenty-two (61.1%) did not use any respiratory protection (*p* = 0.011). The study data showed that occupational exposure to wildfire smoke was associated with the development of a restrictive pattern and associated factors included a sedentary lifestyle, limited duration of physical exercise, longer exposure to fire smoke and non-use of respiratory protection.

## 1. Introduction

Wildfires are one of the most serious natural disasters in the world, not only because of the high frequency in which they occur and the extent they reach, but also because of the destructive effects they cause and Portugal is not excluded from this context.

In Portugal, in 2020, there were 9619 wildfires that burnt a total of 67,170 hectares [1].

A fire department is an operational unit, homologated and technically organized, prepared and equipped for the full exercise of the missions assigned to it [2]. As of 2019, there were 26,939 firefighters and 465 fire departments in Portugal [3,4].

A firefighter is a specially trained individual who, voluntarily or professionally, is integrated in a fire department and has a duty of firefighting, providing prevention and relief services to the community to protect life, environment, and heritage [2]. In the performance of their duties, firefighters are subject to risk factors capable of causing work accidents or catching relevant occupational diseases [5].

The working environment of firefighters is different from that of any other profession, not only because of physical risks but also due to the respiratory and systemic risks of inhaling smoke from combustion, since smoke has in its composition a large number of particles that contain free radicals of hydrogen, carbon, and oxygen [6].

The prevalence of obstructive patterns in a population of firefighters is higher than the national prevalence of chronic obstructive pulmonary disease (COPD), suggesting that exposure to smoke from fires may contribute to the disease [7].

The research available on the firefighter population is very limited and there is insufficient data to conclude the relationship between exposure to smoke and chronic cardiovascular and respiratory effects. Thus, this investigation aimed to determine the respiratory function of firefighters exposed to wildfire smoke during the summer of 2021 and explore the potential associated risk factors.

## 2. Materials and Methods

The present study was descriptive-correlational and cross-sectional, having been approved by the Research in Education and Community Intervention (RECI).

The investigators contacted commanders of fire departments to inform them of the study’s aims and procedures prior to receiving authorization for data collection in their facilities.

All participants signed an informed consent form, having been informed about the nature and purpose of the study, the procedures implicit in the collection of the data for investigation, and the possibility of withdrawal at any time, without any kind of injury.

### 2.1. Population

The study population consisted of firefighters from southern Portugal, of any age and of all genders.

Inclusion criteria cumulatively considered the following factors: firefighters who were working at the time of data collection, who participated in at least one fire between June and September 2021, those who wished to participate voluntarily and who signed the informed consent form. They excluded firefighters with diagnosed respiratory diseases and underwater firefighters.

In the southern region of Portugal, there are 17 fire departments. The choice of the two fire departments (Albufeira and Portimão) was for the convenience of the investigator. In 2021, 68 firefighters performed firefighting, representing the Albufeira corporation and in the Portimão corporation, 90 operational firefighters participated in this fight. This information was provided by their respective commanders.

The sample calculation was determined using the population of 158 firefighters, with an estimated prevalence of respiratory pattern alteration in 12% of the firefighters, verified in a national study [7], with a sample error of 10% and a confidence interval of 99%. Based on these data, the minimum sample size was stipulated at 49 firefighters.

### 2.2. Measuring Instruments

The measurement instruments used were a scale, a stadiometer, a questionnaire, a Fagerstrom test, and a spirometer.

Data collection was carried out in October 2021 at the premises of the fire departments that participated in the study.

### 2.3. Scale and Stadiometer

The SECA 760 digital scale with a capacity of 150 kg and accuracy of 1000 g was used. The stadiometer used has a measuring scale of up to 200 cm.

For the measurements on the scale, the participants remained standing barefoot and with as little clothing as possible. In the measurements with the stadiometer, the participants remained standing with their backs to it, with their heads in the Frankfurt plane.

The values given by the scale and stadiometer were used to calculate the body mass index (BMI). BMI values below 18.5 kg/m^2^ classified the individual as underweight, between 18.5 kg/m^2^ and 24.9 kg/m^2^ with adequate weight, between 25 kg/m^2^ and 29.9 kg/m^2^ with excess weight, and values higher than 30 kg/m^2^ with obesity [8].

### 2.4. Questionnaire

The researchers constructed a questionnaire with closed and multiple-choice questions and applied it in the form of an interview by the researcher.

The questionnaire was subject to consideration by members of the fire department of the population under study, namely the commander and deputy commander of one of the participating fire departments and by a Professor with a Ph.D. in Public Health. Afterwards, a pre-test was performed on 15 firefighters from fire departments who did not participate in the study. In this pre-test, the filling time and the level of understanding were assessed using the Likert Scale. The main filling time was 3.73 min.

The questions focused on age, gender, the act and frequency of physical exercise, smoking habits, number of fires in which they had participated between June and September 2021, time in combat, use of respiratory protection material in combat (wildland firefighter face mask without particle filter, wildland firefighter face mask with FFP2 particle filter and motorized respiratory system), whose response options were as follows: always, usually, occasionally, never; presence of symptoms of smoke inhalation poisoning (nausea, difficulty breathing, headache, drowsiness and disorientation) and resort to medical care. Individuals who claimed to be smokers completed the Fagerstrom test.

### 2.5. Fagerstrom Test

The Fagerstrom test was used to assess nicotine dependence. This test was validated for the Portuguese population by Ferreira et al. [9] and presents test-retest reliability, ensured by correlation values on the original scale of 0.990, and from 0.975 to 1000 in individual questions. The Cronbach alpha coefficient was 0.660.

The Fagerstrom test consists of 6 multiple-choice questions ranging from 0 to 3 points according to the question; questions 1 and 6 range from 0 to 3 points, and the remaining from 0 to 1 point. The sum of quotations allows us to obtain a classification on the degree of nicotine dependence that is translated into very low (0–2), low (3–4), medium (5), high (6–7), and very high (8–10) [9].

### 2.6. Spirometer

Spirometry is a test that helps in the prevention and allows the diagnosis, and quantification, of ventilatory disorders. Respiratory function was evaluated using the open-circuit spirometer Spirodoc Mir with disposable, single-use one-way mouths.

The spirometer used was the Spirodoc mir, which involves open circuit spirometry, is pocket sized, with the interface protocol RS232, precision (wave length, nm) 3% or 50 mL and 5% or 200 mL/s, flow range, 1/s up to 16, flow resistance (plots) < 1 cm H_2_O/L/s; a calibration method was not required.

The spirometer provided the spirometric values and automatically encoded these values in the respiratory pattern categorizations (obstructive, restrictive, mixed type, non-specific, and normal). Respiratory patterns were subcategorized according to the severity of the commitment, according to American Thoracic Society (ATS)/European Respiratory Society (ERS), 2005.

The parameters evaluated were as follows: forced vital capacity (FVC), forced expiratory volume in the first second (FEV1), forced expiratory flow (PEF), medium forced expiratory flow (FEV2575), forced expiration time (FET), and Tiffeneau index percentage (FEV1%) calculated by the formula $\frac{\mathrm{FEV}1}{\mathrm{FVC}}\times 100$.

FVC is the volume of air eliminated in forced expiration from total lung capacity (TLC) to residual volume (RV). TLC is the amount of air present in the lungs after a maximum inspiration and the amount of air in the lungs after maximum expiration is VR. TLC and RV cannot be measured by spirometry [10].

FEV1 is the amount of air eliminated in the first second of forced expiration and is the most clinically useful measurement of respiratory function [10].

PEF or peak flow is the peak expiratory flow reached at forced expiration [11].

FEV2575 expresses the average airflow measured in the interval between 25% and 75% of FVC [11]. The FET represents the total time, in seconds, during which the FVC maneuver occurs [10].

The Tiffeneau index (FEV1/FVC) represents FEV1 in relation to FVC and should be around 68% to 85% of FVC. The classical literature has adopted 80% as the lower limit of normality, and the values below characterize an obstructive ventilatory deficiency [11]. In obstructive pulmonary diseases, such as COPD, the characteristic change in spirometry is the reduction in the Tiffeneau index, which may indicate the presence and severity of airway obstruction [12].

In the obstructive ventilatory pattern, changes are present at the expiratory flow level. Decreased values of variables FEV1, Tiffeneau index, peak-flow and FEV2575 manifest the presence of an obstruction. Table 1 shows the classification on the degree of obstruction based on the values of FVC, FEV1, and FEV1/FVC.

In the restrictive ventilatory pattern, all static volumes are decreased, but there is not necessarily a decrease in flow, especially from the Tiffeneau index [11].

All spirometries were performed by the same investigator (physiotherapist), who explained the entire procedure to each participant. For the spirometric measurement, the participant remained in a standing position, and performed maximal inspiration, followed by a period of apnea; then the mouthpiece connected to the spirometer was introduced, where the participant subsequently performed fast and strong expiration for as long as possible, exhaling the volume of air present in the lungs.

To familiarize the participant with the spirometry procedures, the participant was asked to perform the activity once without the equipment. Then, the participant repeated the procedure with the equipment so that the spirometric parameters considered were taken for the investigation.

### 2.7. Data Analysis

The software Statistical Package for the Social Sciences (SPSS), version 28 for Windows was used to treat the data collected and also for the statistical analysis.

Descriptive statistics analysis was performed using frequency distribution, central trend, and dispersion measures.

To determine the association between the variables used in the study, inferential statistics were performed, with nonparametric tests, namely the chi-square independence test and the level of statistical significance established for α was 5%.

The applicability of the chi-square independence test requires that less than 20% of the expected frequencies are less than 5 [13]. To comply with this criterion of applicability, some variables were grouped. The variable age was grouped into two categories considering the median value (39). Similarly, other variables were grouped into two categories considering the median value, including weekly frequency of physical exercise (3); weekly time of physical exercise (3); the number of wildfires fought (4), and hours in combat (48). BMI was also grouped into two categories, these being “adequate weight” with BMI values equal to or less than 24.9 kg/m^2^ and “overweight” with BMI values equal to or greater than 25 kg/m^2^. In the case of the level of nicotine dependency, two categories were defined, including low dependence (score ≤ 4) and high dependence (score ≥ 5). The variable corresponding to respiratory protection during combat was also grouped into the following two categories: without the use of respiratory protection and with the use of respiratory protection.

## 3. Results

The study sample involved 53 operatives, aged between 23 and 60 years (39.28 ± 8.71), with 41 (77.40%) males, and 12 (22.60%) females.

Thirty (56.60%) firefighters belonged to the Albufeira fire department and twenty-three (43.40%) to Portimão.

The firefighters presented BMIs with values between 17.63 kg/m^2^ and 36.73 kg/m^2^ (27.20 ± 4.15 kg/m^2^). In regards to the BMI classification, only 1 (1.9%) firefighter was classified as underweight, 13 (24.5%) firefighters had adequate weight, 26 (49.1%) overweight, 11 (20.8%) with grade I obesity, and 2 (3.8%) individuals with grade II obesity.

Regarding the practice of physical exercise, 23 (43.4%) firefighters declared not to perform any physical exercise and 30 (56.6%) declared that they work out. The main frequency of physical exercise was 2.97 (standard deviation: 1.07) times a week, with an average weekly time of 4.23 (standard deviation: 2.61) h.

Smoking habits were observed in 28 (52.8%) firefighters. When evaluating nicotine dependence data, it was found that 7 (13.2%) individuals had a very low degree of dependence, 12 (22.6%) revealed a low degree of dependence, 5 (9.4%) had a medium degree and 1 (1.9%) individual had a high degree of dependence.

In the fight against wildfires between June and September 2021, firefighters fought between 1 and 9 (3.64 ± 1.97) fires, with a total of combat time between 5 and 212 (62.34 ± 46.89) h.

In regards to the use of respiratory protection during firefighting, 8 (15.1%) firefighters stated that they did not use any respiratory protection, 18 (34%) firefighters used respiratory protection occasionally, 23 (43.4%) most of the time and 4 (7.5%) firefighters reported using respiratory protection during the entire fight. All 45 (100%) firefighters who used respiratory protection used as equipment the fabric mask with an FFP2 particle filter.

According to the data collected, 49 (92.5%) firefighters denied having symptoms of respiratory intoxication (nausea, difficulty breathing, headache, drowsiness, and disorientation) and 4 (7.5%) reported having only respiratory difficulty as a symptom. Only one (25%) individual reported having to resort to medical care.

Table 2 presents the descriptive statistics values of the evaluated spirometric parameters.

The results collected by spirometry revealed that 17 (32.1%) individuals presented normal values and 36 (67.9%) individuals presented changes in the ventilatory pattern, as shown in Table 3. None of the individuals presented an obstructive ventilatory pattern or a mixed pattern (restrictive/obstructive).

The association between the various variables analyzed in the present study and the presence of the alteration of the restrictive ventilatory pattern are represented in Table 4.

To investigate the association between the fire departments and the presence of restrictive ventilatory patterns, the relationship between the variables and the fire department under study was carried out, as shown in Table 5.

## 4. Discussion

Data from the present study showed the presence of a restrictive ventilatory pattern in most firefighters (68%) whom performed fire fighting between June and September 2021. Lower values were found in the study by Almeida et al. [7] that evaluated 203 firefighters who participated in the fight against forest fires in Portugal between December 2005 and January 2006 and their data revealed the prevalence of an obstructive pattern in 11.8% of the firefighters.

Although in the present study, the presence of an obstructive ventilatory pattern was not observed, it was observed that, in comparison to the study of Almeida et al. [7], the percentage that revealed a change in the ventilatory pattern was higher. This can be hypothetically explained by the different moments of data collection. Data collection in the present study occurred no later than 3 months after the first exposure to smoke from wildfires, whereas in the study by Almeida et al. [7], the collection was carried out at the end of the year, with a longer gap.

The measurement instruments used and the ventilatory parameters collected also differed between these studies, which could explain the fact that the obstructive pattern was absent in the present study and in the study by Almeida et al. [7], the restrictive pattern was absent. In the study by Almeida et al. [7], the Piko-6^®^ spirometer was used and the spirometric parameters collected for the evaluation of the ventilatory pattern were only FEV1 and the maximum expiratory volume at 6 s (FEV6). The ratio between these (FEV1/FEV6) represents an alternative method for the identification of bronchial obstruction, in relation to the usual FEV1/FVC ratio considered in the present study, which allows the identification of restrictive and obstructive ventilatory patterns.

The longitudinal study of Mathias et al. [14] between 2009 and 2016 evaluated changes in pulmonary function over time in 662 firefighters (with occupational routine firefighting) through 2 separate evaluations for 5 years, where they evaluated the same spirometric parameters as the present study (FVC, FEV1, and FEV1%) and observed a reduction in respiratory function and FEV1 and FVC (percentage of predicted) decreased (*p* < 0.001) from 100.9 ± 0.6% to 92.3 ± 0.5% and 99.0 ± 0.6% to 91.9 ± 0.5%, respectively, concluding that changes in the ventilatory pattern in the firefighters studied were two to four times greater (*p* < 0.001) than the changes expected over 5 years.

The variables in the study that indicated that they were statistically associated with the presence of restrictive ventilatory patterns were the practice of physical exercise, the amount of weekly physical exercise, the time in combat, and the use of respiratory protection.

The presence of sedentary habits among the firefighters analyzed in this study was high (43%), and the presence of restrictive ventilatory patterns was observed in 53% of these individuals. Of the individuals who had a restrictive ventilatory pattern and also practiced physical exercise (70.6%) did so on a weekly basis for equal to or less than 3 h, which is considered a low amount of training to obtain physical fitness gains [15]. By analyzing this data, it was found that a sedentary lifestyle and low duration of physical exercise were potential factors for the presence of a restrictive ventilatory pattern. The weekly frequency of training was not relevant.

In the present study, the number of fires fought was not associated with the presence of a restrictive ventilatory pattern; however, the amount of time of exposure to smoke showed a positive association, where 56% of the firefighters who battled for more than 48 h presented a restrictive ventilatory pattern. Thus, greater exposure to smoke from wildfires resulted in a potential risk factor for the presence of changes in the respiratory pattern.

The study by Serra et al. [16] examined the respiratory function of 92 firefighters, whose main activity was wildfire firefighting and a control group of 51 police officers and they also did not find a positive association between respiratory function and the number of fires. Exposure time was not evaluated in this study.

In the light of the results obtained, wildfire firefighting cannot be considered a risk-free activity for respiratory function. Wildfires produce smoke with abundant particles in the inhalable particle size range (<100 μm) and contain carbon-free radicals and precursors [17] resulting from the combustion of plant material.

In the results obtained on the use of respiratory protection, it was observed that 61.1% of the firefighters with changes in the ventilatory pattern did not use respiratory protection, data similar to those obtained in the study by Almeida et al. [7], who found that 95.8% of firefighters with obstructive patterns did not use protective airway material. The respiratory protection material considered in this study consisted of the use of the wildland firefighter face mask with an FFP2 particle filter and the study by Almeida et al. [7] focused on the flash hood as the validated respiratory protection, which consists of thermal protection equipment.

The firefighters studied, whom reported using respiratory protection in firefighting (100%), used the same type of protection (wildland firefighter face mask with FFP2 particle filter); thus, it was not possible to check which respiratory protection is most effective. The different respiratory protection materials for wildfire fighting are wildland firefighter face masks with or without FFP2 particle filters, which work under negative pressure and require an additional respiratory effort from the firefighter, who sometimes opts not to use it, in order to save efforts. Alternatively, there is a monitored respiratory system with the use of HEPA P3 particle filter that works at positive pressure, does not require additional respiratory effort and presents greater particle filtration [18].

Although a statistically significant association was not verified between smoking habits and the presence of restrictive ventilatory patterns, most firefighters with changes in the respiratory pattern (53%) displayed smoking behaviors. Similar data were observed in the study by Almeida et al. [7], where a positive association was not observed between these variables, even though 58% of firefighters with obstructive patterns demonsrated smoking habits. Jacquin et al. [19] studied the changes in the spirometric parameters FVC, FEV1, and FEV1% during the wildfire season, from June to September, in 108 firefighters. This study compared a group of smokers (59) with a non-smoker group (49) and found no statistical association for the parameters in this comparison.

Since there was a statistically significant association between the Albufeira and Portimão fire departments (*p* = 0.006), we chose to carry out a more detailed analysis considering these fire departments to understand the reason for this association. The data obtained revealed that the firefighters of the Portimão corporation practiced more weekly hours of physical exercise compared to the firefighters of the Albufeira corporation (69% versus 29%). Thus, it is suggestive that this association between fire departments and the presence of a restrictive ventilatory pattern is justified by the duration of physical exercise.

This study presented some limitations, namely the calculation of the sample not having included a greater number of fire departments in the southern region of Portugal, and the fact that only a small percentage of the firefighters of the corporations analyzed had been fighting fires. The rest of the operational body was involved in other activities. Another limitation was expressed in the absence of questions about the time in which firefighters belonged to the active firefighting body and coronavirus disease (COVID-19). Finally, the absence of spirometric values at a time before the start of the wildfire season to compare the results after the fighting also consisted a limitation. This type of study only allows us to explore the strength and direction of the relationships that exist between certain variables, verifying the way that a phenomenon is accompanied by the appearance of another phenomenon. Thus, it is suggested to conduct longitudinal studies that are characterized by the study of causal relationships and also future studies that involve a more representative sample of the population of firefighters at a national level.

## 5. Conclusions

The data from this study allowed us to verify that occupational exposure to smoke from wildfires was associated with the development of a restrictive ventilatory pattern in firefighters and that the factors associated with this presence included sedentary lifestyle, reduced duration of physical exercise, longer exposure time to fire smoke and non-use of respiratory protection material.

It is important to perform periodic screenings for respiratory diseases in this type of population, to promote awareness campaigns to encourage regular physical exercise and the use of respiratory protection material, as well as the acquisition of respiratory systems motorized in positive pressure as respiratory protection.

## Figures and Tables

**Table 1 ijerph-19-08492-t001:** Classification on the degree of obstruction based on FVC, FEV1 and FEV1/FVC values [11].

Degree	FVC % of the Expected	FEV1 % of the Expected	FEV1/FVC % of the Expected
Slight	60 ^1^	60 ^1^	60 ^1^
Moderate	51–59	41–59	41–59
Serious	50	40	40

^1^ Lower limit.

**Table 2 ijerph-19-08492-t002:** Descriptive statistics of the evaluated spirometry parameters.

Parameter	Average ± Standard Deviation	Min–Max
FVC (L)	3.29 ± 0.98	1.27–5.43
FEV1 (L)	3.10 ± 0.87	1.27–5.22
FEV1% (%)	96.36 ± 5.95	72.2–100
PEF (L/s)	8.07 ± 1.98	3.51–11.97
FEV2575 (L/s)	5.28 ± 1.45	2.69–9.26
FET (s)	1.22 ± 0.80	0.36–4.34

**Table 3 ijerph-19-08492-t003:** Spirometry results: classification of the ventilatory pattern.

Classification of Spirometry	Absolute Frequency	Relative Frequency (%)
Normal lung function	17	32.10
Obstructive ventilatory pattern	0	0
Restrictive ventilatory pattern (slight restriction)	16	30.20
Restrictive ventilatory pattern (moderate restriction)	8	15.10
Restrictive ventilatory pattern (moderately severe restriction)	6	11.30
Restrictive ventilatory pattern (severe restriction)	6	11.30
Mixed type (obstructive and restrictive)	0	0
Non-specific	0	0
TOTAL	53	100.0

**Table 4 ijerph-19-08492-t004:** Association between variables and the ventilatory pattern.

Variables		Restrictive Ventilatory Pattern		*p*-Value
		Absent	Present	
Fire Department	Albufeira	5 (29.4%)	25 (69.4%)	0.006
	Portimão	12 (70.6%)	11 (30.6%)	
Gender	Male	15 (88.2%)	26 (72.2%)	0.194 ^1^
	Female	2 (3.8%)	10 (27.8%)	
Age	≤39 years	8 (47.1%)	20 (55.6%)	0.563
	>39 years	9 (52.9%)	16 (44.4%)	
BMI	Adequate weight	7 (41.2%)	7 (19.4%)	0.094 ^1^
	Overweight	10 (58.8%)	29 (80.6%)	
The practice of physical exercise	No	4 (23.5%)	19 (52.8%)	0.045
	Yes	13 (76.5%)	17 (47.2%)	
Weekly frequency of exercise practice	≤3 times a week	8 (61.5%)	14 (82.4%)	0.201 ^1^
	>3 times a week	5 (38.5%)	3 (17.6%)	
Weekly time of physical exercise	≤3 h	4 (30.8%)	12 (70.6%)	0.030
	>3 h	9 (69.2%)	5 (29.4%)	
Smoking habits	No	11 (64.7%)	17 (47.2%)	0.234
	Yes	6 (35.3%)	19 (52.8%)	
Level of nicotine dependence	Low dependency	4 (66.7%)	15 (78.9%)	0.539 ^1^
	High dependency	2 (33.3%)	4 (21.1%)	
Number of wildfires fought	≤4 wildfires	14 (82.4%)	22 (61.1%)	0.122
	>4 wildfires	3 (17.6%)	14 (38.9%)	
Time in combat	≤48 h	13 (76.5%)	16 (44.4%)	0.029
	>48 h	4 (23.5%)	20 (55.6%)	
Use of respiratory protection	Without the use	4 (23.5%)	22 (61.1%)	0.011
	With the use	13 (76.5%)	14 (38.9%)	

^1^ did not comply with the conditions of applicability of the chi-square independence test.

**Table 5 ijerph-19-08492-t005:** Relationship between the variables and the fire departments of Albufeira and Portimão.

Variables		Fire Departments	
		Albufeira	Portimão
Gender	Male	22 (73.3%)	19 (82.6%)
	Female	8 (26.7%)	4 (17.4%)
Age	≤39 years	16 (53.3%)	12 (52.2%)
	>39 years	14 (46.7%)	11 (47.8%)
BMI	Adequate weight	9 (30%)	5 (21.7%)
	Overweight	21 (70%)	18 (78.3%)
The practice of physical exercise	No	13 (43.3%)	10 (43.5%)
	Yes	17 (56.7%)	13 (56.5%)
Weekly frequency of exercise practice	≤3 times a week	13 (76.5%)	9 (69.2%)
	>3 times a week	4 (23.5%)	4 (30.8%)
Weekly duration of physical exercise	≤3 h	12 (70.6%)	4 (30.8%)
	>3 h	5 (29.4%)	9 (69.2%)
Smoking habits	No	15 (50.0%)	13 (56.5%)
	Yes	15 (50.0%)	10 (43.5%)
Level of nicotine dependence	Low dependency	10 (66.7%)	9 (90.0%)
	High dependency	5 (33.3%)	1 (10.0%)
Number of wildfires fought	≤4 wildfires	20 (66.7%)	15 (68.2%)
	>4 wildfires	10 (33.3%)	7 (31.8%)
Time in combat	≤48 h	17 (76.5%)	11 (50.0%)
	>48 h	13 (43.3%)	11 (50.0%)
Use of respiratory protection	Without the use	13 (43.3%)	13 (59.1%)
	With the use	17 (56.7%)	9 (40.9%)
Restrictive ventilatory pattern	Absent	5 (16.7%)	12 (52.2%)
	Present	25 (83.3%)	11 (47.8%)

## Data Availability

The data obtained in this study are included in an SPSS database. Informed consent and the tests applied are on paper, filed at our institution. None of these documents are available online, only in the paper analyzed.

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
