# Peer review of "Prevalence and Factors Associated with Poor Respiratory Function among Firefighters Exposed to Wildfire Smoke"

_ijerph, 2022, doi:10.3390/ijerph19148492_

Round 1

Reviewer 1 Report

The authors report a study of respiratory health of wildland firefighters in southern Portugal. It is an important occupational health issue and their finding that many firefighters do not use respiratory protection is a major concern. The work certainly has merit but it is very difficult to read. A thorough English-language edit is needed. Other edits for clarity are listed below:

Line 8: Is this any rural fire or specifically forest or wildfires? Wildfire is the preferred term. Awkward sentence. Suggest rewording - "Wildfires are an increasingly common type of disaster." 

Line 12: Add 'years of age'

Line 13: Use text for numbers that begin a sentence - "Thirty-six"

Line 34: Re-word to eliminate repetition. "The firefighter is a specially trained individual, who ...

Line 51: specify year

Line 55: Term "Blinded Information" should not be used. Some descriptive text must be developed to use in the manuscript or leave it out/delete. Or re-word this. For example: "The present study was approved after appropriate institutional review for human subjects protections. The study analyses are descriptive-correlational and cross-sectional."

Lines 56-57: Again awkward sentence. Suggest this instead: "Investigators contacted commanders of fire brigades to inform them of the study aims and procedures prior to receiving   authorization for data collection in their facilities."  

Line 63: Simplify the sentence: The study population consisted of firefighters from southern Portugal, of any age and of all genders.

Line 70: "firefighter divers"

Line 71: Is it fire brigades or fire departments? choose one term and use it consistently.

Line 72: If the information is blinded just don't include it. Suggest: "Two (of 17) fire brigades were selected for the convenience of the investigators. One brigade had 68 eligible firefighters and the other had 90 eligible firefighters."

Line 98: Delete "Since questionnaires are not known to evaluate the theme studied" and begin the sentence with "The researchers"

Line 107: Again, simplify. "The questions included focused on..."

Line 164: This is confusing. Are you talking about familiarizing the participant? If so, re-wording is recommended. "To familiarize the participant with the spirometry procedures, the participant was asked to perform the activity once without the equipment. Then, the participant repeated the procedure with the equipment so that the ..."  

Line 178: "This measure was also considered to the group" This is confusing. I think you mean this: "Similarly, other variables were grouped into two categories considering the median value: weekly frequency of physical exercise (4)..."

Lines 191-192: Delete sentence with blinded information.

Line 194: Delete skilled.

Table 4: Remove "Blinded Information"

Line 232-233 and Table 5. If you need to distinguish between each fire department, please give each a generic label, e.g., Department 1 or Department A. Delete "Blinded Information"

Table 5: Change units of weekly time of physical exercise to hours.

Line 288: I am not familiar with 'inland range'. Do you mean inhalable or respirable?

Line 304: Rather than the term 'Opposite' it is better to use "Alternatively". 

Line 316-317: A statistically significant association between what variable and fire department? Also, please use a generic label to identify each fire department.

Author Response

Thank you for affording us another opportunity to make our article read better for the global audience. We appreciate all the comments and suggestions that were made by two reviewers. I hope we have made the revisions to your satisfaction.

REVIEWER 1:

The authors report a study of respiratory health of wildland firefighters in southern Portugal. It is an important occupational health issue and their finding that many firefighters do not use respiratory protection is a major concern. The work certainly has merit but it is very difficult to read. A thorough English-language edit is needed. Other edits for clarity are listed below:

Answer: The language was proofread by a translator.

Line 8: Is this any rural fire or specifically forest or wildfires? Wildfire is the preferred term. Awkward sentence. Suggest rewording - "Wildfires are an increasingly common type of disaster." 

Answer: In Portugal forest fires are called rural. However, it makes sense to make this change for forestry. Thanks.

Line 12: Add 'years of age'

Answer: Done.

Line 13: Use text for numbers that begin a sentence - "Thirty-six"

Answer: Done.

Line 34: Re-word to eliminate repetition. "The firefighter is a specially trained individual, who ...

Answer: Done.

Line 51: specify year

Answer: Done.

Line 55: Term "Blinded Information" should not be used. Some descriptive text must be developed to use in the manuscript or leave it out/delete. Or re-word this. For example: "The present study was approved after appropriate institutional review for human subjects protections. The study analyses are descriptive-correlational and cross-sectional."

Answer: The term "Blinded Information" was used to eliminate any information that could identify the authors, in order to guarantee a blind review. We already put the names.

Lines 56-57: Again awkward sentence. Suggest this instead: "Investigators contacted commanders of fire brigades to inform them of the study aims and procedures prior to receiving   authorization for data collection in their facilities."  

Answer: Done.

Line 63: Simplify the sentence: The study population consisted of firefighters from southern Portugal, of any age and of all genders.

Answer: Done.

Line 70: "firefighter divers"

Answer: Done.

Line 71: Is it fire brigades or fire departments? choose one term and use it consistently.

Answer: We choose departments.

Line 72: If the information is blinded just don't include it. Suggest: "Two (of 17) fire brigades were selected for the convenience of the investigators. One brigade had 68 eligible firefighters and the other had 90 eligible firefighters."

Answer: We already put the names of the departments.

Line 98: Delete "Since questionnaires are not known to evaluate the theme studied" and begin the sentence with "The researchers"

Answer: Done.

Line 107: Again, simplify. "The questions included focused on..."

Answer: Done.

Line 164: This is confusing. Are you talking about familiarizing the participant? If so, re-wording is recommended. "To familiarize the participant with the spirometry procedures, the participant was asked to perform the activity once without the equipment. Then, the participant repeated the procedure with the equipment so that the ..."  

Answer: Done.

Line 178: "This measure was also considered to the group" This is confusing. I think you mean this: "Similarly, other variables were grouped into two categories considering the median value: weekly frequency of physical exercise (4)..."

Answer: Done.

Lines 191-192: Delete sentence with blinded information.

Answer: Done.

Line 194: Delete skilled.

Answer: Done.

Table 4: Remove "Blinded Information"

Answer: Done.

Line 232-233 and Table 5. If you need to distinguish between each fire department, please give each a generic label, e.g., Department 1 or Department A. Delete "Blinded Information"

Answer: We already put the names of the departments.

Table 5: Change units of weekly time of physical exercise to hours.

Answer: We changed to: Weekly duration of physical exercise

Line 288: I am not familiar with 'inland range'. Do you mean inhalable or respirable?

Answer: We put this: “Wildfires fires produce fumes with abundant particles in the inhalable particle size range (<100μm) and contain carbon-free radicals and precursors [17] resulting from the combustion of plant material.”

Line 304: Rather than the term 'Opposite' it is better to use "Alternatively". 

Answer: Done.

Line 316-317: A statistically significant association between what variable and fire department? Also, please use a generic label to identify each fire department.

Answer: “a statistically significant association between the Albufeira and Portimão fire departments”. We put the name of departments.

Reviewer 2 Report

Title: The impact of exposure to rural fire fumes on the respiratory 2 function of firefighters

Comments

General comments

This is an important study that investigates firefighting as an occupation with risk of lung diseases. This is not a much-studied area yet this information could provide information on prevention measures for lung diseases in the fire fighters.

This manuscript would benefit from editorial support to strengthen the language and writing.

Specific comments

Title: This was a cross sectional study, and this design helps to identify associations rather than impact. Impact is best determined using prospective cohort studies

The study methods including approach to data analysis are not described in the abstract.

The introduction provides a lot of information about textbook facts on fire fighting. This study is about fire fighters and association with lung diseases, I would therefore expect the introduction to focus on that. For example, what is the burden of lung diseases in general, in Portugual, etc. What is the burden of lung diseases attributable to exposure to fires among firefighters? Is there a difference between fires in the urban and rural?

What are the implications?

The rationale for this study is also not provided.

Methods

The study participants were from 2 fire departments, and both have exactly the same name ‘BLINDED INFORMAT’ON'. It is difficult to make sense of this and gets more confusing in the presentation of the results. I propose that the unique identifiers are added- maybe geographical location.  In addition, the authors analysed data from the 2 departments, indicating that there are potential differences in the 2 fire departments that could influence lung health in the fire fighters. For this reason, these study sites must be described more in the context of the study objective.

The authors need to provide more details on the study procedures, especially spirometry, instead of providing basic facts about it. For example,

·       What equipment was used?

·       Who conducted the tests?

·       How was quality assurance done?

·       Was there any screening of the participants for relative contra-indications to Spirometry?

·       Which standards were used? ERS/ATS guidelines?

·       Were any quality checks done on the Spirograms?

·       Was reversibility testing done?

·       What happened to the individuals with abnormal results?

The grading of the use of respiratory protection needs to be described further. What does ‘occasionally, most of the time, etc refers to? At least an estimate of the frequency of use in numbers should be provided.

The spirometry results need more description. What references were used in interpretation of the results? ERS/ATS? Please describe on how the categorization was done. The basic categorizations are; obstructive, restrictive, mixed type, non-specific, and normal.

What criteria were used to determine the cut-offs that were used to categorize age, frequency of exercise, hours of exercise and hours of combat, and provide the references or rationale. Were these categorization based on any scientific evidence in relation to the subject under study?

Lines 188-291 present the demographic and clinical characteristics of the participants. This information is very important in understanding the sample. I propose that the authors consider presenting this is a table for easy of understanding.

The grading of restrictive disease is done using TLC. The results in table 3 provide information about the severity of the restrictive ventilatory defect in this population. Was this based on TLC?

The pathological changes in the lungs that manifest as ventilatory defects usually occur over a period of time. It would have been appropriate to collect information about the years of service as a fire fighter to provide a sense of length of exposure. This is a limitation in this study. This is also in relation to the conclusions made in line 339-340 indicating that the results in this study which investigated exposure over a few months was associated with development of restrictive disease. There are many other possible causes of restrictive disease, some having occurred in childhood, that could be responsible for the findings. These possibilities need to be discussed.

If screening for lung health is done prior to recruitment, this too can provide very valuable information on the role of exposure to fires and lung health among fire fighters.

Author Response

Thank you for affording us another opportunity to make our article read better for the global audience. We appreciate all the comments and suggestions that were made by two reviewers. I hope we have made the revisions to your satisfaction.

REVIEWER 2

General comments

This is an important study that investigates firefighting as an occupation with risk of lung diseases. This is not a much-studied area yet this information could provide information on prevention measures for lung diseases in the fire fighters.

This manuscript would benefit from editorial support to strengthen the language and writing.

 Answer: The language was proofread by a translator.

Specific comments

Title: This was a cross sectional study, and this design helps to identify associations rather than impact. Impact is best determined using prospective cohort studies

Answer: Thanks for the comment which I totally agree with. The title and purpose of the study were changed. “Thus, this investigation aimed determine the respiratory function in firefighters exposed to wildfires fire fumes in the last summer of 2021 and explore potential associated risk factors,”

The study methods including approach to data analysis are not described in the abstract.

Answer: If we put this information we will exceed the number of words in the abstract.

The introduction provides a lot of information about textbook facts on fire fighting. This study is about fire fighters and association with lung diseases, I would therefore expect the introduction to focus on that. For example, what is the burden of lung diseases in general, in Portugual, etc. What is the burden of lung diseases attributable to exposure to fires among firefighters? Is there a difference between fires in the urban and rural?

Answer: I preferred not to include these data in the introduction section precisely because we think it will take the reader's focus away from the main objective of our study.

what is the weight of lung diseases in general, in Portugal, etc. - We will not study lung diseases in the general population, but the respiratory changes in firefighters who were exposed to fire smoke and we put this phrase in the introduction section “The prevalence of obstructive pattern in a population of firefighters is higher than the national prevalence of Chronic Obstructive Pulmonary Disease (COPD)”

What is the burden of lung disease attributable to fire exposure among firefighters? - Few studies have this information and we prefer to use these results to discuss with the data obtained in our study and not be presenting them in the introduction section.

The objective was not to verify the prevalence of pulmonary diseases.

Is there a difference between urban and rural fires? - the focus is on rural fires, which smoke differs from urban fire, and this is not part of the study. If I explain this difference the reader may think that we will investigate this.

What are the implications?

The rationale for this study is also not provided.

Answer: Because fire is a major problem worldwide and also in Portugal, and because there are few studies investigating the association between exposure to smoke and respiratory problems, this was the way we found to justify our study.

“Wildfires fires are one of the most serious natural disasters in the world, not only because of the high frequency with which they occur and the extent they reach, but also because of the destructive effects they cause, and Portugal is not excluded from this context.

In Portugal, in 2020, there were 9619 wildfires fires with a total of 67,170 hectares of burned area [1].”

“The research available on the firefighter population is very limited and there is insufficient data to conclude the relationship between exposure to fumes and chronic cardiovascular and respiratory effects”

Methods

The study participants were from 2 fire departments, and both have exactly the same name ‘BLINDED INFORMAT’ON'. It is difficult to make sense of this and gets more confusing in the presentation of the results. I propose that the unique identifiers are added- maybe geographical location.  In addition, the authors analysed data from the 2 departments, indicating that there are potential differences in the 2 fire departments that could influence lung health in the fire fighters. For this reason, these study sites must be described more in the context of the study objective.

Answer: The term "Blinded Information" was used to eliminate any information that could identify the authors, in order to guarantee a blind review. We already put the names.

We do not agree that these study sites should be described more in the context of the objective of the study as it is not part of the main objective. Since there were differences between the departments, we decided to carry out one more analysis to try to justify this difference.

The authors need to provide more details on the study procedures, especially spirometry, instead of providing basic facts about it. For example,

  • What equipment was used?

Answer: We added this phrase: “The spirometer used was the Spirodoc mir, pocket sized, interface protocols RS232, precision (wave length, nm) 3% or 50 ml and 5% or 200 mL/sec, flow range, 1/sec up to 16, flow resistance (Plots) <1 cm H2O/L/sec, calibration method not required.”

  • Who conducted the tests?

Answer: We added this: “All spirometry was performed by the same investigator (Physiotherapist),”

  • How was quality assurance done?

Answer: We do not carry out quality assurance in these types of assessment. I'm not familiar with this procedure.

  • Was there any screening of the participants for relative contra-indications to Spirometry?

Answer: According to our exclusion criteria: “They excluded firefighters with diagnosed respiratory diseases and divers firefighters.”

  • Which standards were used? ERS/ATS guidelines?

Answer: We added this paragraph: “The spirometer provided the spirometric values and automatically encoded these values in the respiratory pattern categorizations (obstructive, restrictive, mixed type, non-specific, and normal). Respiratory patterns were subcategorized according to the severity of the commitment, according to American Thoracic Society (ATS)/European Respiratory Society (ERS), 2005.”

  • Were any quality checks done on the Spirograms?

Answer: The spirometer did not provide spirograms. The results appeared on the spirometer display.

  • Was reversibility testing done?

Answer: No. We used a open circuit spirometry.

  • What happened to the individuals with abnormal results?

 Answer: The abnormal results would be restriction or obstruction, and these data are presented in our results.

The grading of the use of respiratory protection needs to be described further. What does ‘occasionally, most of the time, etc refers to? At least an estimate of the frequency of use in numbers should be provided.

 Answer: We added this information in the methods section: “use of respiratory protection material in combat (wildland firefighter face mask without particle filter, wildland firefighter face mask with FFP2 particle filter and motorized respiratory system) whose response options were always, mostly, occasionally, never”

The spirometry results need more description. What references were used in interpretation of the results? ERS/ATS? Please describe on how the categorization was done. The basic categorizations are; obstructive, restrictive, mixed type, non-specific, and normal.

Answer: We added this paragraph: “The spirometer provided the spirometric values and automatically encoded these values in the respiratory pattern categorizations (obstructive, restrictive, mixed type, non-specific, and normal). Respiratory patterns were subcategorized according to the severity of the commitment, according to American Thoracic Society (ATS)/European Respiratory Society (ERS), 2005.”

What criteria were used to determine the cut-offs that were used to categorize age, frequency of exercise, hours of exercise and hours of combat, and provide the references or rationale. Were these categorization based on any scientific evidence in relation to the subject under study?

Answer: We chose to use the median value, as described. This option was due to the fact that we were able to meet the applicability conditions of the statistical test.

Lines 188-291 present the demographic and clinical characteristics of the participants. This information is very important in understanding the sample. I propose that the authors consider presenting this is a table for easy of understanding.

Answer: The article already has 4 tables in the resuts section, we would not like to add one more.

The grading of restrictive disease is done using TLC. The results in table 3 provide information about the severity of the restrictive ventilatory defect in this population. Was this based on TLC?

Answer: The spirometer did not provide data of TLC.

“The parameters evaluated were: forced vital capacity (FVC), forced expiratory volume in the first second (FEV1), Forced Expiratory Flow (PEF), Medium Forced Expiratory Flow (FEV2575), Forced Expiration Time (FET), and Tiffeneau Index percentage (FEV1%) calculated by the formula: .”

“In the restrictive ventilatory pattern, all static volumes are decreased, but there is not necessarily a decrease in flow, especially from the Tiffeneau Index [11].”

The pathological changes in the lungs that manifest as ventilatory defects usually occur over a period of time. It would have been appropriate to collect information about the years of service as a fire fighter to provide a sense of length of exposure. This is a limitation in this study.

Answer: The years of service were not questioned because in Portugal firefighters can have many years of profession, but not have participated in many wildfires fires, for example they could have 20 years of profession, but they only fought fires in 3 years. We think it's better to question about the number of fires that participated between June and September 2021. Furthermore, our objective was to verify the association of short-term exposure to fire fumes shortly after 3 months of exposure. The objective was not to verify the prevalence of pulmonary diseases.

This is also in relation to the conclusions made in line 339-340 indicating that the results in this study which investigated exposure over a few months was associated with development of restrictive disease. There are many other possible causes of restrictive disease, some having occurred in childhood, that could be responsible for the findings. These possibilities need to be discussed.

Answer: There are other factors that could lead to the development of respiratory changes that were not controlled in the study, as they are not completely controlled in several other studies. We controlled for some variables such as smoking, physical activity, excluded those with diagnosed diseases, but we did not control for factors that could have occurred in childhood, so we think it makes no sense to discuss them. The objective was not to verify the prevalence of pulmonary diseases.

If screening for lung health is done prior to recruitment, this too can provide very valuable information on the role of exposure to fires and lung health among fire fighters.

Answer: Is this recruitment you refer to before individuals perform the role of firefighters? We did not understand.

We refer this fact as a limitation of our study. “This study presented some limitations, namely the calculation of the sample not having included a greater number of fire departments in the southern region of Portugal, and the fact that only a small percentage of the firefighters of the quartiles analyzed had been fighting fires. The rest of the operational body was involved in other activities. Another limitation was expressed in the absence of questions about the time in which firefighters belonged to the active firefighting body and coronavirus disease (Covid-19). Finally, the absence of spirometric values at a time before the start of the season of wildfires fires, to compare the results after the fighting, also consisted a limitation.”

Round 2

Reviewer 2 Report

Comments

Title: The ‘risk factors’ is hanging. Risk factors for what? You can either leave this out and this would give you room to describe the respiratory function or if you want to highlight the associated factors, you could consider a title like ‘Prevalence and factors associated with poor respiratory function among firefighters….’

Abstract

Line 10:  ..and explore potential risk factors associated. This is incomplete. Associated with ….Title:

Results: Logical flow can be better to highlight the key findings in this study. My understanding is that the authors aimed to understand the proportion of wild firefighters with poor lung function, and related factors. It is important to note that when describing lung function based on Spirometry- the following have to be mentioned;

·       Normal lung function

·       Obstructive ventilatory defect/pattern

·       Restrictive ventilatory defect/pattern

·       Mixed type (obstructive and restrictive)

·       Non-specific

When providing information on factors associated with poor lung function, it is better to simply state that fighters who had less than …hours of exercise and …..hours of exposure to fires were more likely to have poor lung function (OR, CI, p value), instead of tagging this to a particular ventilatory defect. The was it is stated now would seem like the authors wanted to find out the factors associated with specific ventilatory defects ( eg restrictive, obstructive, etc).

Introduction

The authors do a good job of describing the burden of fires in Portugal. However, I miss the information on the burden of poor respiratory function in Portugal, and how this may compare with that in fire fighters. This information would provide clear rationale for studying this group.

There is need to describe the risk factors for poor lung function in this group (apart from exposure to smoke) and point out studies done elsewhere.

In the concluding statement, the part on ‘to raise awareness and implement more appropriate protective measures’ was not studied and should not be included as a study aim. It should be stated as a way in which the results might be used.

Methods

The information in line 131-135 shows that the authors used ATS/ERS standards to categorize the ventilatory defects. However, the reference for table 1 (ref 15) is a textbook by the American College of Sports Medicine. ACSM’s Guidelines for Exercise Testing and Prescription. (10th). Wolters Kluwer. 2017. Could the authors give reasons why the used the ones from American College of Sports Medicine to classify the degree of obstruction? The relevance of this table in the methods section is also not very clear, especially because there were no participants with obstructive defects.

In the discussion, p values should not be included. Instead, state the interpretation of the p values.

Author Response

Thank you for reviewing our article again. Some requests were unclear, we tried to respond, other suggestions we did not agree with and no changes were made and other suggestions, although we did not think they were adequate, changes were made as requested. I hope that we were able to respond to the request.

Comments

Title: The ‘risk factors’ is hanging. Risk factors for what? You can either leave this out and this would give you room to describe the respiratory function or if you want to highlight the associated factors, you could consider a title like ‘Prevalence and factors associated with poor respiratory function among firefighters….’

Answer: Changed: “Prevalence and factors associated with poor respiratory function among firefighters exposed to wildfire smoke”

Abstract

Line 10:  ..and explore potential risk factors associated. This is incomplete. Associated with ….Title:

Answer: Added: This study aims to determine the respiratory function on firefighters exposed to wildfire smoke and explore potential risk factors asssociated with poor respiratory function.

Results: Logical flow can be better to highlight the key findings in this study. My understanding is that the authors aimed to understand the proportion of wild firefighters with poor lung function, and related factors. It is important to note that when describing lung function based on Spirometry- the following have to be mentioned;

  • Normal lung function
  • Obstructive ventilatory defect/pattern
  • Restrictive ventilatory defect/pattern
  • Mixed type (obstructive and restrictive)
  • Non-specific

Answer: We do not understand. Do you want to include in the table values that were not obtained and in the text before the table it is stated that no obstructive changes were found? If so, we made this change to the table, although I don't think it's adequate.

Classification of spirometry

Absolute frequency

Relative frequency (%)

 Normal lung function

17

32,10

Obstructive ventilatory pattern

0

0

Restrictive ventilatory pattern - slight restriction

16

30,20

Restrictive ventilatory pattern -moderate restriction

8

15,10

Restrictive ventilatory pattern -moderately severe restriction

6

11,30

Restrictive ventilatory pattern - severe restriction

6

11,30

Mixed type (obstructive and restrictive)

0

0

Non-specific

0

0

TOTAL

53

100,0

When providing information on factors associated with poor lung function, it is better to simply state that fighters who had less than …hours of exercise and …..hours of exposure to fires were more likely to have poor lung function (OR, CI, p value), instead of tagging this to a particular ventilatory defect. The was it is stated now would seem like the authors wanted to find out the factors associated with specific ventilatory defects ( eg restrictive, obstructive, etc).

 Answer: We did not perform logistic regression, so we cannot present these data.

Introduction

The authors do a good job of describing the burden of fires in Portugal. However, I miss the information on the burden of poor respiratory function in Portugal, and how this may compare with that in fire fighters. This information would provide clear rationale for studying this group.

Answer: We insist that it makes no sense for us to include this information because we do not study lung diseases in the general population, but the respiratory changes in firefighters who were exposed to fire smoke. We cannot compare data from a general population with a specific group of professionals who have their own work characteristics that will influence health standards.

There is need to describe the risk factors for poor lung function in this group (apart from exposure to smoke) and point out studies done elsewhere.

Answer: The research available on the firefighter population is very limited, so there's not much to discuss.

In the concluding statement, the part on ‘to raise awareness and implement more appropriate protective measures’ was not studied and should not be included as a study aim. It should be stated as a way in which the results might be used.

 Answer: The part about “raising awareness and implementing more appropriate protection measures” was removed as suggested, but had not been included as an objective of the study, the objective was described below as a way of justifying the importance of carrying out this study, that is, indicating a way in which the results could be used.

Methods

The information in line 131-135 shows that the authors used ATS/ERS standards to categorize the ventilatory defects. However, the reference for table 1 (ref 15) is a textbook by the American College of Sports Medicine. ACSM’s Guidelines for Exercise Testing and Prescription. (10th). Wolters Kluwer. 2017. Could the authors give reasons why the used the ones from American College of Sports Medicine to classify the degree of obstruction?

Answer: It was a mistake. The correct reference is number 11.

The relevance of this table in the methods section is also not very clear, especially because there were no participants with obstructive defects.

Answer: But we had no way of predicting that there would be no obstructive patterns. This was the methodology applied, but we did not find other changes that were not restrictive. Therefore the table must be maintained.

In the discussion, p values should not be included. Instead, state the interpretation of the p values.

Answer: The p-values were removed as per your suggestion, but could be presented this way in the discussion; it's another way of revealing, without writing, that there was statistical significance.

“The variables in the study that indicated that they were statistically associated with the presence of restrictive ventilatory pattern were the practise of physical exercise, the amount of weekly physical exercise, the time in combat, and the use of respiratory protection.”